# An Integrated Optical Circuit Architecture for Inverse-Designed Silicon Photonic Components

**DOI:** 10.3390/s23020626

**Published:** 2023-01-05

**Authors:** Dusan Gostimirovic, Richard Soref

**Affiliations:** 1Department of Electrical and Computer Engineering, McGill University, Montreal, QC H3A 0G4, Canada; 2Engineering Department, University of Massachusetts at Boston, Boston, MA 02125, USA

**Keywords:** silicon photonics, inverse design, optical switches, optical networks

## Abstract

In this work, we demonstrate a compact toolkit of inverse-designed, topologically optimized silicon photonic devices that are arranged in a “plug-and-play” fashion to realize many different photonic integrated circuits, both passive and active, each with a small footprint. The silicon-on-insulator 1550-nm toolkit contains a 2 × 2 3-dB splitter/combiner, a 2 × 2 waveguide crossover, and a 2 × 2 all-forward add–drop resonator. The resonator can become a 2 × 2 electro-optical crossbar switch by means of the thermo-optical effect, phase-change cladding, or free-carrier injection. For each of the ten circuits demonstrated in this work, the toolkit of photonic devices enables the compact circuit to achieve low insertion loss and low crosstalk. By adopting the sophisticated inverse-design approach, the design structure, shape, and sizing of each individual device can be made more flexible to better suit the architecture of the greater circuit. For a compact architecture, we present a unified, parallel waveguide circuit framework into which the devices are designed to fit seamlessly, thus enabling low-complexity circuit design.

## 1. Introduction

Silicon photonics has experienced remarkable growth in cost-effective applications including optical computing [1,2,3], sensing [4,5], and optical communications [6,7,8]. The silicon photonics platform leverages the modern CMOS foundry fabrication infrastructure to yield fully integrated electronic–photonic wafers and chips. Photonic inverse design has been introduced recently as a method of accelerating and automating the design of advanced chips [9,10].

A survey of the photonics literature shows that inverse optical design is a research-and-development area that has emerged strongly during the past five years. This design literature reports both theoretical and experimental results. We can identify specific developments that are relevant to the work reported in this paper. These include the computational techniques of inverse design that are assisted by deep neural networks [11], deep learning [12,13], physics-informed machine learning [14], black box algorithms [15], integral equation methods [16], and linkage tree genetic algorithms [17]. Additional computational methods include photonic emulation [18], the deep adjoint approach [19], phase-injected topology optimization [20], and boundary integral methods [21].

Practical implementation of inverse-designed components in commercial silicon photonics foundries has been discussed by authors who examined spatial process variations [22,23], structural integrity [24], foundry fabrication constraints [25], and 300 mm multi-project wafer fabrication [26]. Progress on inverse-design nanophotonic components has been reported in the relevant areas of on-chip microresonators [27], planar on-chip mode sorters [17], all-optical logic devices [28], fiber-to-chip metamaterial edge couplers [29], optical beam steerers [30], couplers for on-chip single-photon sources [31], polarization splitter-rotators [32], and photonic arbitrary beam splitters [33]. On the network level, inverse-design developments include neural networks [34], Stokes receivers [35], and integrated photonic circuits [36].

We have extended the foregoing prior art to the silicon photonic inverse-design area. Silicon-on-insulator (SOI) photonic integrated circuits (PICs) are constituted of waveguided active and passive components. Some components have become ubiquitous, such as directional couplers, waveguide crossovers, and microring resonators (MRRs) that are side coupled to bus waveguides. Others, important but less pervasive, are star couplers, mode converters, N × M multimode interferometers, 1 × N splitters, and N × 1 combiners. Generally, all of these can be created using the inverse-design methodology presented in this paper. Each of the components have issues of large footprint or large optical crosstalk, or, for the MRR, the second bus feeds backward instead of forward. Such issues “motivate” the inverse-designed component (IDC) approach. The IDC approach investigated here can resolve many of these issues, but at the expense of increased fabrication complexity. To create IDCs, the high resolution photolithography of the foundry is required to define a pixelated array of tiny air holes (later filled with oxide) within an initially uniform ~220 nm silicon rectangular film that connects input strip waveguides to the output strip waveguides. This photoetching of silicon is performed in one processing step for all IDCs in the circuit. We speculate that the tradeoff in complexity will be worthwhile because of the enhanced component performance.

The investigation presented in this work focuses on three IDCs, each having two input ports and two output ports: a 3 dB coupler (splitter/combiner), a 100% coupler (crossover), and a new square micro-resonator with feed-forward behavior. Electro-optical crossbar switching of the square resonator (also known as reconfiguration of an add–drop multiplexer) is simulated here by means of a uniform change in the real index of the pixelated silicon film, accomplished, for example, by the thermo-optical (TO) effect. Examples of these IDCs working together cooperatively in PICs are presented. This paper focuses on the SOI platform operating at the 1550 nm telecom wavelength; however, our IDC approach applies to any “on-insulator” semiconductor platform, which means any group IV, III-V, or II-VI semiconductor circuit on the oxidized silicon wafer substrate. In the context of wavelength-division multiplexed on-chip systems, we present passive and active (switched) IDC circuits. Ten specific circuit applications are illustrated in this paper.

## 2. The Inverse-Design Approach

The literature reports IDC results in coarse wavelength-division (de)multiplexers (CWDM) [10,37], mode converters [38,39], bends [40], and waveguide crossings [41,42] with performance per unit area far greater than that of conventional, hand-based designs. Such design is initiated by the desired objective function (e.g., maximizing modal throughput from one input waveguide to another output waveguide) and by defining a highly dimensional design space (often in the form of a “pixelated” permittivity matrix) to be modified by the inverse-design algorithm. Perhaps the most successful of these inverse-design algorithms is topology optimization [43,44], which iteratively defines the hundreds to thousands of material permittivity pixels in the design area to maximize the specified objective function—producing highly nonintuitive designs that are highly unlikely to be achieved by hand design. Optimization of such a highly dimensional design problem is made possible by the adjoint method [45,46], which requires only two simulations to be made per optimization iteration, where the design variables (permittivity pixel distribution) are ultimately updated based on the gradient of their effect on the objective function.

For system-level design, where individual components may have to share key parameters and fabrication process-specific design constraints, inverse design is a useful method, as the skeleton of the framework can be generally defined by the designer and the blanks (the device design areas) can be filled in to suit the desired functionality, performance targets, and design constraints. In this work, we present a unified framework for easily arrangeable photonic circuits built with inverse-designed components. This framework is a series of parallel waveguides with fixed spacing, but a variable number of waveguides depending on the function and scale of the circuit. For example, a monochromatic 8 × 8 crosspoint matrix switch uses 16 parallel waveguides, and a monochromatic 8 × 8 Spanke–Benes permutation matrix switch uses eight. Having set the framework, a set of “plug-and-play” building block devices can be arranged within it to carry out the desired functionality.

In this work, we present three inverse-designed, topologically optimized SOI photonic building blocks: the 2 × 2 3-dB splitter/combiner, the 2 × 2 waveguide crossover, and the 2 × 2 all-forward add–drop resonator. Each of these devices feature two parallel waveguides with the same size and spacing to match the circuit framework, and a topologically optimized design region within the center to carry out the optical functionality. To demonstrate the flexibility of this approach, we present the designs of ten different circuits.

We aim to create a simplified framework for PICs to promote design uniformity, short design times, low complexity, low spatial footprint, high optical performance, and high signal quality. To achieve these goals, the core components (devices) use the highest quality design methods, and the unifying framework must be set up to support these devices with a structure that suits them best. We believe the method of topological inverse design produces the best performing devices; given the parallel input/output structure generally used with the algorithm, we believe a parallel series of waveguides best supports them while keeping circuit complexity low. The design of the circuit and each individual component follow the same “fill-in-the-blank” process: for the devices that have a high degree of design dimensionality, we delegate the design work to a computer algorithm; for the circuit, which has a low degree of dimensionality, we perform the design work by hand.

## 3. The Inverse-Designed Components

The 2 × 2 3-dB splitter/combiner, 2 × 2 crossover, and 2 × 2 all-forward add–drop resonator presented in this section are designed with topology optimization using the adjoint method from the open-source topology optimization software, Angler [46], where a 2.5D effective-index FDFD solver is used for each device simulation. For this work, the software was modified to enable multi-objective optimizations. For the optimization of the first two devices, we initially considered two parallel waveguides spaced 2 μm apart (forming two input waveguides and two output waveguides) and an inverse-design region in the center of the two parallel waveguides. This would demonstrate small footprint while achieving high optical performance. The required size and waveguide spacing is larger for the resonator, however, and so for consistent integration of components, we then created a new splitter/combiner and a new crossover whose spacings matched those of the resonator (10 × 10 μm).

The optimizations are set up for each device to maximize a target objective function with respect to its permittivity distribution. Target objective functions are typically a maximizing of optical throughput for a given mode and wavelength from an input waveguide to an output waveguide. The inverse-design region in the middle of each device is discretized into a matrix of 40 nm pixels that can each take a permittivity value of silicon (ε_r,Si_ = 12.1) or silica (ε_r,SiO_2__ = 2.07) based on the direction the optimizer takes in maximizing the objective function. To efficiently optimize the thousands of pixels in the design region, topology optimization employs the adjoint method to calculate the optimization gradient, as it only requires two simulations per iteration to calculate. The first simulation is a typical simulation of the device; the second simulation swaps the input and output directions to produce a “reverse simulation” to obtain the gradient of the objective function with respect to the set of design parameters (pixel permittivity distribution). With the gradient calculated, an L-BFGS-B optimizer [47] adjusts the permittivity distribution along the gradient by a predetermined step size towards a slightly better performing design. This procedure is repeated until the objective function can no longer be maximized (fully converged). The overall method has shown success in designing compact high-performance silicon photonic devices and is easily adaptable to new devices with unique design elements and functionalities, such as ours.

### 3.1. 2 × 2 3-dB Splitter/Combiner

The 2 × 2 3-dB splitter/combiner, as shown in Figure 1, is the essential passive component of Mach–Zehnder interferometer switches. Its functionality is such that a signal at either of the two input waveguides is split with a 50:50 ratio at the two output waveguides, ideally with minimal loss and back reflection. Should a same-wavelength signal be present at each input, the two combine at one of the two output ports, depending on the relative phase of the two signals. The device features the standard, 220 nm SOI platform, with 500 nm-wide input and output waveguides spaced 9 μm apart. A dimension of 10 × 10 μm^2^ is chosen for the inverse-design region (square component in the middle of the device) to match the other two devices presented later in this work.

The topological optimization is set up to maximize 50:50 splitting for a 1550 nm TE_0_ mode, whether it comes in at the top input waveguide or the bottom input waveguide. Although the devices in this work are optimized only for TE_0_ (i.e., they are not polarization-insensitive), they can just as readily be re-optimized for TM_0_. For polarization-insensitive operation, both polarizations can be optimized simultaneously through an extra condition in the objective function; however, this would likely occur at the expense of peak performance. The combiner functionality is not explicitly included in the optimization, as we can expect complete reciprocity of the splitter function in this passive optical device.

A large low-pass spatial filter (commonly applied to modern topologically optimized devices [48]) is applied to finalize a design that contains typically larger features that are fabricated more reliably. The filter used smooths out features with length scales of less than 10 pixels (400 nm). In general terms, a smaller low-pass spatial filter produces designs with smaller features that are more difficult to fabricate, but also allows for higher performance, as a larger design space can be explored in optimization. We carefully manage this tradeoff to avoid extremes.

Figure 1 shows the top-view optical field profiles of the device for each expected initial condition (other than no input being present). A 50:50 (top:bottom) split is visualized for the first two initial conditions.

The exact splitting ratios at 1550 nm are determined to be 0.466:0.490 and 0.456:0.472 for top-waveguide and bottom-waveguide inputs, respectively, which indicate low insertion loss (0.20 dB and 0.32 dB). For the two initial conditions that satisfy the combiner function, as shown in Figure 2, the phase of the two inputs determine which output port the signals combine at. For phase conditions A (inputs have a π/2 phase difference) and B (inputs have a −π/2 phase difference), the combining ratios are 1.840:0.042 and 0.003:1.882, respectively, which indicate low insertion loss (0.36 dB and 0.26 dB) and low crosstalk (−16.8 dB and −28.2 dB).

Like that of the nonintuitive inverse-design region, the optical field profile within follows a pattern that is difficult to decode. We observe that the signals take a nonlinear path from the input to the output, employing the entire inverse-design region, but remain well confined within it. Given the extremely high dimensionality of the optimization problem, it is unsurprising that the performance and complexity of the final device design region are both high.

For cases using two splitter/combiner devices, there is a simple IDC procedure to create a high-performance 2 × 2 Mach–Zehnder interferometer crossbar switch. First, two parallel waveguides are used to connect the two outputs of the first splitter to the two inputs of the second combiner. Then, an electro-optical phase shifter (EOPS) is inserted into one of the two connecting waveguide arms. The bar state of the 2 × 2 is a stable non-volatile state where the EOPS is zero. The cross state is reached when EOPS has π radians of shift.

### 3.2. 2 × 2 Crossover

The 2 × 2 crossover, as shown in Figure 3, is the second key passive component of higher order integrated optical switches. Its functionality is such that a signal at either of the two input waveguides will output at the diagonally opposite port (cross state), with minimal loss and back reflection. This functionality is perhaps more like an evanescent waveguide coupler than a traditional waveguide crossing, which features perpendicular inputs/outputs [41] that would not easily fit our parallel waveguide circuit architecture. The inverse-designed crossover has an advantage over conventional evanescent waveguide couplers, as the complex (inverse) index engineering creates new effective paths so that full coupling can occur for virtually any device length (including very short lengths). Like the 2 × 2 3-dB splitter/combiner, this device features the standard 220 nm SOI platform with 500-nm-wide input and output waveguides spaced 9 μm apart. The same compact dimension of 10 × 10 μm^2^ is chosen for the design region. The two devices presented thus far appear to be similar, but the highly dimensional design regions are distributed differently.

The topology optimization is set up to maximize power transfer from the top input waveguide to the bottom output waveguide, and from the bottom input waveguide to the top output waveguide, for a 1550 nm TE_0_ mode. To maintain similarity with the 2 × 2 3-dB splitter/combiner, the same large spatial filter is also applied here to obtain a final design that contains the same easy-to-fabricate large features. Figure 3 shows the top-view optical field profiles of the device for each of the two previously mentioned initial conditions, where we observe almost complete waveguide crossing for both. The exact crossing ratio is 0.003:0.959 (top:bottom) and 0.911:0.001 for optical stimulation at the top and bottom input waveguides, respectively, which indicates low insertion loss (0.18 dB and 0.40 dB) and low crosstalk (−25 dB and −29 dB).

The potential advantages and benefits of the proposed crossover and splitter/combiner devices can be seen by comparing these designs to the current art of “traditional” splitters and crossovers. This comparison can be made in four areas: (1) the spectral transmission bandpass, (2) the device footprint, (3) the insertion loss, and (4) the optical crosstalk. The traditional 2 × 2 3-dB splitters comprise the classical directional coupler (CDC) (two, identical, parallel strip waveguides with an identical S-bend waveguide at each of four ports), cascaded DCs, the MMI coupler, and the asymmetric coupler with four tapers [49]. The Figure 1 inverse-designed splitter shares with the CDC a narrow bandpass of 4 nm for 0.15 dB of output port imbalance, but the inverse device has a much smaller footprint, and its IL and XT metrics are superior. The traditional crossover is generally an X-shaped structure with waveguides oriented at right angles to each other, and with several waveguiding shapes being practical [41]. Our analysis of the Figure 3 crossover shows that the spectral bandpass (4 nm for 0.2 dB of IL imbalance) is smaller than that of traditional designs; however, the Figure 3 design—whose footprint is much similar in area to the traditional designs—offers convenient parallelism of the four waveguides, unlike the traditional 90-degree orientation. The design offers IL and XT comparable to those of traditional crossings.

### 3.3. 2 × 2 All-Forward Add–Drop Resonator

The 2 × 2 all-forward add–drop resonator, as shown in Figure 4, is the third key passive component. It is intended to be the functional equivalent of the two-bus-coupled MRR, but here it takes a monolithic shape of silicon without evanescent coupling. Like the MRR, an off-resonance signal at the top left input waveguide will exit at the top right output waveguide (through port); but different from the MRR, the resonance signal here exits at the bottom right output (drop port) instead of the bottom left. To the best of our knowledge, this is the first gap-less all-forward 2 × 2 resonator—a device designed by topology optimization for high levels of performance in a compact spatial footprint.

Like the other two devices in this toolkit, this device features the standard 220 nm SOI platform with 500 nm-wide input and output waveguides spaced 9 μm apart (vertically). To achieve good performance (i.e., high extinction ratio and quality factor), the design region is set to 10 × 10 μm^2^. In general terms, topologically optimized devices perform better than their conventionally designed counterparts per the same unit area. Although this device is two times larger than the “smallest” 2 × 2 MRRs, the all-forward arrangement is a complex constraint for the optimization and more room must be provided for the optical signal to diverge from its natural “resonant path” and exit in the forward direction.

The topological optimization is set up to maximize power transfer from the top left input waveguide to one of the two right-side output waveguides in such a way that light is maximized at the drop port (bottom output) for 1550 nm and is maximized at the through port (top output) for 1548 nm and 1552 nm (again, for a TE_0_ mode). This objective function guides the optimizer to create a resonator with a large extinction ratio for a quality factor of approximately 4500, as shown in the Lorentzian-like transmission spectra in Figure 4d. Like photonic crystal nanocavities, this device features only one peak, which carries benefit in high-throughout, multichannel circuits. A larger quality factor of the peak can be achieved with a tradeoff in optimization time, extinction ratio, or feature size/complexity. Likewise, a smaller quality factor can improve the other metrics. The transmission plot also shows the spectra for the through and drop ports for a shift in real refractive index of the silicon-square of Δ*n* = 0.003, (a resonance shift of two linewidths), which indicates the device’s ability to achieve full crossbar switching under electro-thermal control, for example.

Figure 4 also shows the optical field profiles from the top view of the device, for each of the two previously mentioned initial conditions. In Figure 4c, the resonance-enhanced buildup of light is clearly shown near the middle of the design region, before exiting diagonally at the drop port. For the off-resonance condition shown in Figure 4f, the optical signal takes a nonintuitive path partially into the design region but is redirected back to the top, through port. This nonlinear path is likely a cause for the through-port (bar state) insertion loss and can likely be reduced with further tuning of the optimization length and parameters.

## 4. Photonic Switch Layouts

The following ten circuits are examples of integrated photonic switches composed of the same core building blocks presented in the previous section. Each circuit also features the same parallel waveguide architecture, where individual devices are placed in a “plug-and-play” fashion to realize unique functionalities in a unified design strategy. By (inverse) designing the core building blocks to be compact, with low insertion loss and low crosstalk, these large, complex circuits can be realized simply and efficiently, ultimately bringing large performance improvements to the chip.

### 4.1. Monochromatic 8 × 8 Crosspoint Matrix Switch

The term monochromatic means that the user chooses a particular wavelength of operation λ_0_ for all signals. After that, the 2 × 2 resonators are designed so that the resonance wavelength λ_r_ matches λ_0_. We assume here that each resonator is an individually, electrically controlled 2 × 2 switch, and, to achieve switching, we can identify three EO resonance-shifting mechanisms: (1) the TO effect in the silicon design region square, where Δ*n* is triggered by a very nearby nano-scale electrical “Joule heater”; (2) the phase change material approach in which a PCM cladding (such as Sb_2_Se_3_) on the silicon square top surface has its phase changed by nano-heaters; or (3) lateral injection of free electrons and holes into the intrinsic silicon square by forward bias applied across P- and N-doped regions at opposite edges of the square.

The classical N x N switch is the “crossbar”, a matrix comprised of 2 × 2 crosspoints (here resonant ones), and in our parallel waveguide approach, we can attain this classical geometry without using any waveguide crossovers. Our design is presented in Figure 5, and here we can illustrate the matrix operation for the TO case by asserting that all 2 × 2s are initially in the cross state, which is the unheated ambient state. Here, the rows and columns are slanted, and we use coincident row–column addressing of one crosspoint in each row to produce a bar state in each row. In Figure 5 and the subsequent Figures, the electrical control wires are not shown for simplicity. The x–y addressing means that, out of the N^2^ devices, only N are addressed at any one time.

### 4.2. Monochromatic 8 × 8 Spanke–Benes Permutation Matrix Switch

In the matrix art, it is known that there are architectures for N inputs and N outputs that have fewer switches than the crossbar does. One such example is the Spanke–Benes permutation matrix switch, requiring only 28 elements for 8 × 8 switching, as we show in Figure 6. Here, again, we attain this matrix without using any waveguide crossovers.

### 4.3. Monochromatic 8 × 8 PILOSS Topology Matrix Switch

For a large-scale addressed matrix in which light travels through various paths, a path-independent insertion loss is a desirable attribute, and for this constraint, the PILOSS topology has been invented [50]. Our next IDC circuit example is based upon the block diagram given in Figure 6c of [50], and here 64 resonator switches and 49 crossovers are deployed for the 8 × 8 PILOSS topology, as presented in Figure 7. Like the other circuits, a uniform parallel array of waveguides is the framework.

### 4.4. Monochromatic Non-Blocking 16 × 16 Clos–Benes Spatial Routing Switch

In the photonics literature, the non-blocking Clos–Benes architecture is an important approach to large-scale routers because the number of elemental switches required is quite low compared to the number in other approaches. However, there is a drawback, which is the “perfect shuffle interconnections” that are required within the matrix, and these consist of a complex array of waveguide crossings at oblique angles. We have invented here a simple triangular array of IDC crossings that perform the needed shuffle, and this method is shown in Figure 8 for the 16 × 16 routing case. Note that here, the number of EO switches is reduced to only 40. A fairly large number of passive crossovers (narrow blocks) is required, as indicated.

### 4.5. Monochromatic 8 × 1 and 1 × 8 Spatial Routing Switch

Some applications require active N × 1 or 1 × N routing, and for this we have a simple approach that is presented in Figure 9. It is interesting that no crossovers are needed for this N = 8 case. The 8 × 1 concept is to route a selected input while blocking the other seven input signals. The small purple squares in Figure 9 represent absorbers or waveguide “terminators”.

### 4.6. 8 × 1 and 1 × 8 Passive Wavelength-Division Multiplexer

At this point, we move into the domain of wavelength-multiplexed circuits and networks. Here, we use a group of 2 × 2 resonators, and we require that each resonator is “dedicated” to a particular λ_r_ or “color”, with that color being different from all other colors in the group. In our diagrams, we “paint” a color on each passive add–drop resonator to indicate its λ_r_, such as red, orange, green, blue, violet, navy, or sienna. The new design is presented in Figure 10 for multiplexer (MUX) and demultiplexer (DEMUX). In this design, it is assumed that an add–drop is in cross state for its color and is in the bar state for any other color. For this MUX/DEMUX, the add–drops are passive.

### 4.7. 2 × 2 × 3λ Multi-Crossbar Switch

Now let us add EO resonance shifting (electrical control) to each wavelength-dedicated 2 × 2 add–drop. A simple-but-effective switch is the cascade connection of three different 2 × 2 switches, as is shown in Figure 11a. This is a wavelength-multiplexed crossbar switch, with each 2 × 2 color “element” being independently controlled. The lower diagram shows the composite spectrum of the device, illustrating individual TO shifts for red, green, and blue. As discussed in [52], a given color input is on resonance for one element (the cross state there) and is off resonance for all other elements (the bar state there). A given element can be shifted from the cross state to the bar state. This multi-crossbar approach is illustrated in Figures 3 and 4 of [52].

### 4.8. 6 × 6 × 4λ Wavelength-Selective Switch

Having just described the operation of the 2 × 2 × Nλ crossbar, let us now put these devices to work in a larger-scale M × M × Nλ wavelength-selective routing switch for the case of 6 × 6 × 4λ, which means that that there are four “color signals” at each of the six inputs and re-arranged (switched) four color signals at each of the four outputs. Our new design is given in Figure 12, where we also deploy multiple 2 × 2 crossovers (blue squares here). This diagram is a highly modified version of the monochromatic 6 × 6 utilizing MRRs presented in [53], where one modification is the replacement of MRRs with our four-color crossbars. If we visualize this as a 24 × 24 router, then the number of active and passive components employed to accomplish the wavelength routing (48 and 9, respectively) is relatively small.

### 4.9. 8 × 8 × 3λ Wavelength-Selective Switch

Our next wavelength-selective switch has eight input channels and eight output channels, each containing three “color signals.” Starting from the monochromatic 8 × 8 in Figure 5 of [54], we expanded that to the wavelength-MUXed application by employing our above Figure 11 device to invent a “24 × 24” router that employs only 60 active devices, as shown in Figure 13.

### 4.10. 4 × 4 × 4λ Wavelength Cross-Connect Switch

The MRR-based “switch and select” approach for wavelength cross-connect switching was recently explored [55], but that approach demands a complicated “shuffle-like” waveguide interconnection. Here, we have embodied the switch-and-select architecture with IDCs, not MRRs, for our last EO PIC proposal. However, we shall deviate from the assumption of an everywhere-parallel waveguide framework because the multi-oblique interconnection cannot be readily attained with IDC crossover arrays, and so the 4 × 4 × 4λ cross-connect switch illustrated here is a “hybrid” that includes the parallel framework and well as a “multi-crossed waveguides” region. Figure 14 presents this wavelength cross-connect switch, which has 48 active devices and 16 passive devices to route 12 incoming WDM signals to 12 outgoing WDM signals. In the parallel region, no crossovers are needed. This nonblocking switch offers low crosstalk.

## 5. Conclusions

We have deployed a unified design strategy to create a compact collection of inverse-designed, topologically optimized, integrated silicon-photonic devices: a 10 × 10 μm^2^ 2 × 2 3-dB splitter/combiner, a matching 2 × 2 crossover, and a matching 2 × 2 all-forward add–drop resonator. The passive add–drop resonator can be converted into an electrically controlled 2 × 2 crossbar switch, where the proposed EO mechanisms are the TO effect, the phase-change-cladding effect (actuated via a nano-heater), or free-carrier injection via lateral PIN structure. The unified shape, input–output positioning, and sizing allow these IDCs to be easily arranged in a simplified, parallel waveguide circuit architecture to perform many different switching and computational functionalities—of which we demonstrate ten different examples in this work in the context of WDM chips. Because of the high efficiency of topological optimization with the adjoint method, extremely high-dimensional design spaces can be explored to create highly performing devices within constricted spatial footprints, such as ours. In other words, by leveraging the power of photonic inverse design, the complexity of circuit design can be significantly reduced while also improving performance. Low levels of insertion loss and of crosstalk are achieved for each of the three devices presented from our toolkit, which ultimately enables larger-scale, lower-power, and higher-bandwidth PICs for next-generation communications and computing applications.

## Figures and Tables

**Figure 1 sensors-23-00626-f001:**
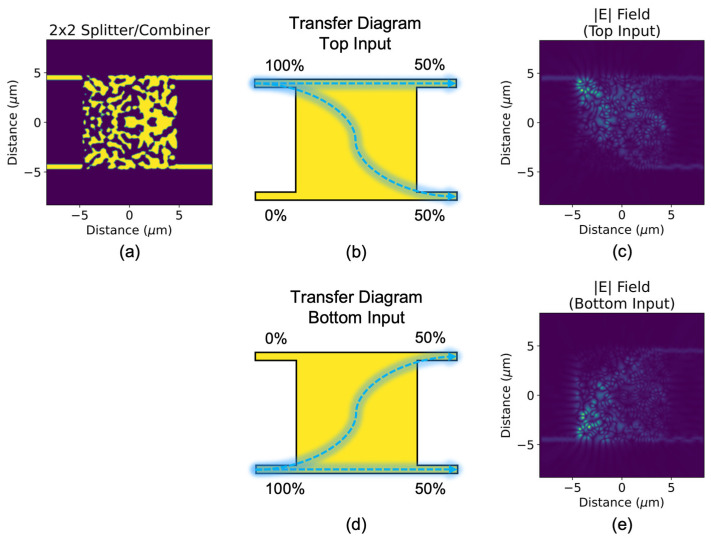
(**a**) Topologically optimized 2 × 2 3-dB splitter/combiner (in splitter mode) and its transfer diagrams for the following initial conditions: (**b**) input at the top waveguide, (**d**) input at the bottom waveguide. Top-view optical field profiles of (**b**,**d**) are shown in (**c**,**e**), respectively.

**Figure 2 sensors-23-00626-f002:**
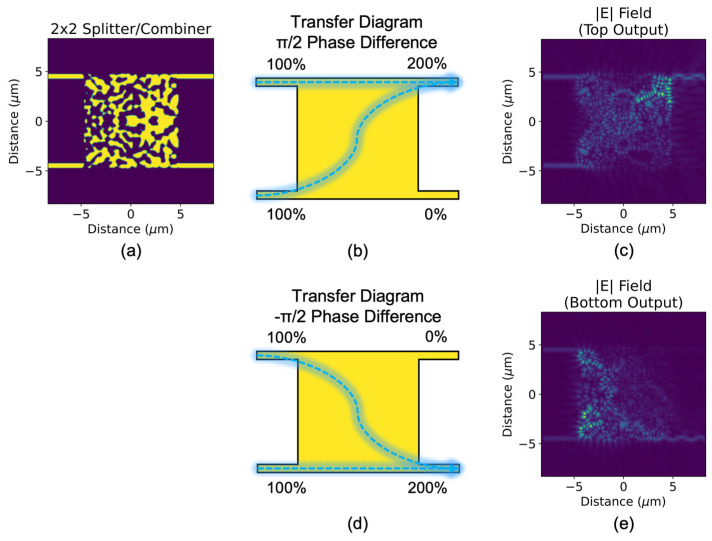
(**a**) Topologically optimized 2 × 2 3-dB splitter/combiner (in combiner mode) and its transfer diagrams for the following initial conditions: (**b**) inputs at both waveguides with π/2 phase difference, (**d**) inputs at both waveguides with −π/2 phase difference. Top-view optical field profiles of (**b**,**d**) are shown in (**c**,**e**), respectively.

**Figure 3 sensors-23-00626-f003:**
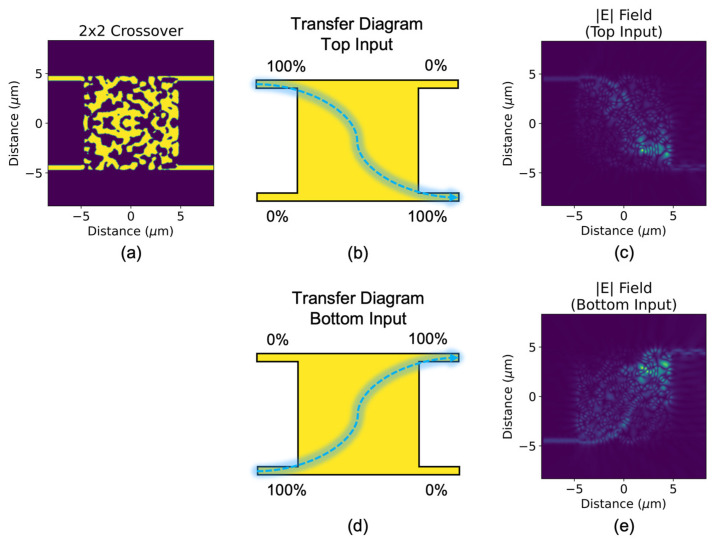
(**a**) Topologically optimized 2 × 2 3-dB crossover and its transfer diagrams for the following initial conditions: (**b**) input at the top waveguide, (**d**) input at the bottom waveguide. Top-view optical field profiles of (**b**,**d**) are shown in (**c**,**e**), respectively.

**Figure 4 sensors-23-00626-f004:**
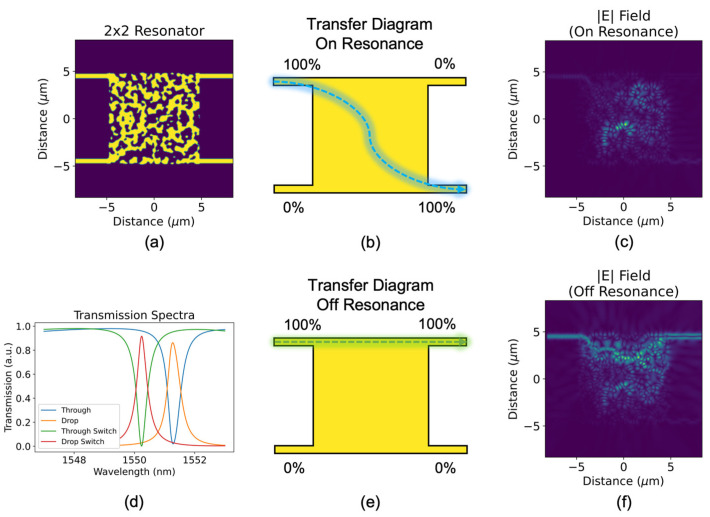
(**a**) Topologically optimized all-forward add–drop resonator, (**d**) its transmission spectrum, and its transfer diagrams for the following initial conditions: (**b**) through-port output (off resonance), (**e**) drop-port output (on resonance). Top-view optical field profiles of (**b**,**e**) are shown in (**c**,**f**), respectively.

**Figure 5 sensors-23-00626-f005:**
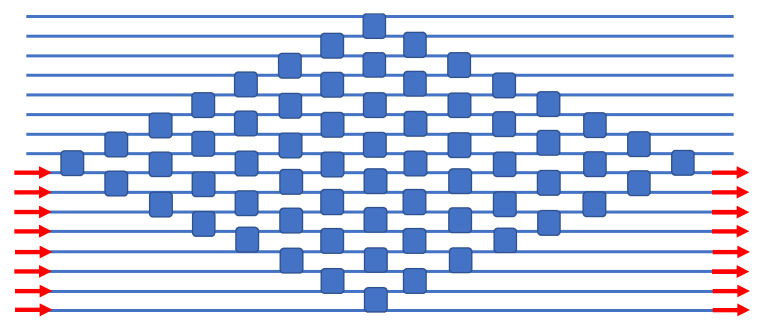
Block diagram of the monochromatic 8 × 8 crosspoint matrix switch. Each blue square indicates an electrically tunable resonant add–drop IDC.

**Figure 6 sensors-23-00626-f006:**
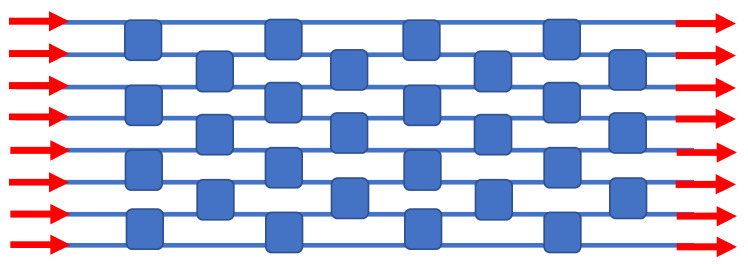
Block diagram of the monochromatic 8 × 8 Spanke–Benes permutation matrix switch. This diagram is based upon the layout of Figure 6b in [50].

**Figure 7 sensors-23-00626-f007:**
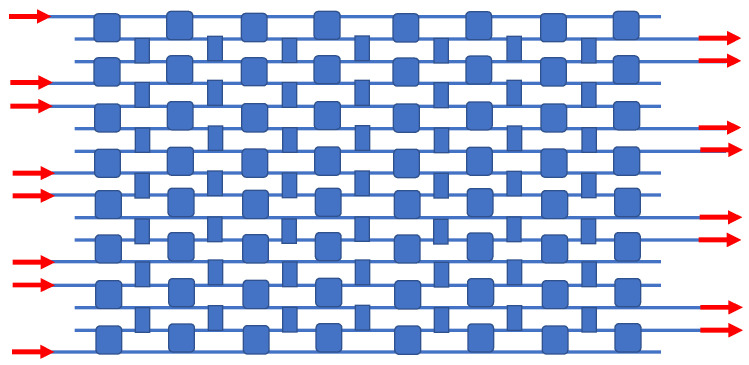
Block diagram of the monochromatic 8 × 8 PILOSS-topology matrix switch. Each narrow blue rectangle represents a passive IDC crossover, with the squares indicating EO switches. This diagram is based upon the diagram in Figure 6c of [50].

**Figure 8 sensors-23-00626-f008:**
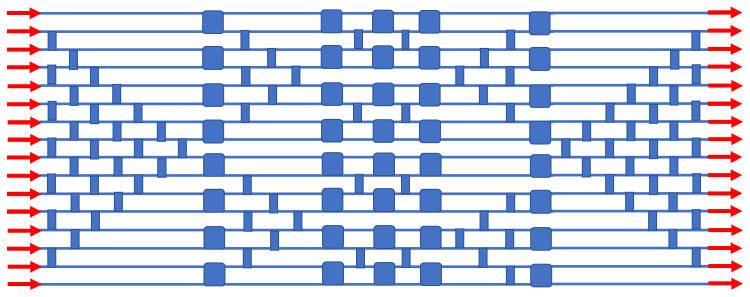
IDC block diagram of the monochromatic non-blocking 16 × 16 Clos–Benes spatial routing switch. This topology is based on the schematic in Figure 4 of [51].

**Figure 9 sensors-23-00626-f009:**
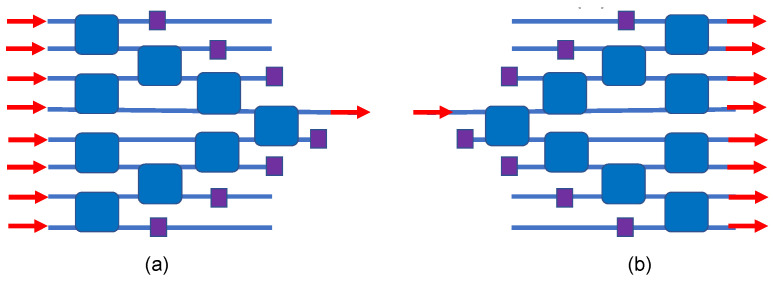
IDC block diagram on monochromatic (**a**) 8 × 1 and (**b**) 1 × 8 spatial routing switches.

**Figure 10 sensors-23-00626-f010:**
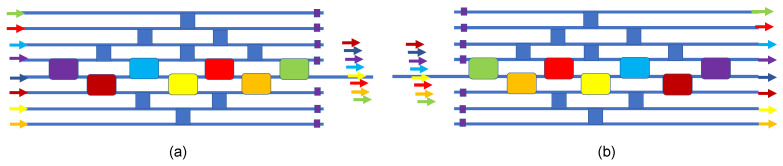
Block diagrams of the wavelength-division (**a**) 8 × 1 multiplexer and (**b**) 1 × 8 demultiplexer.

**Figure 11 sensors-23-00626-f011:**
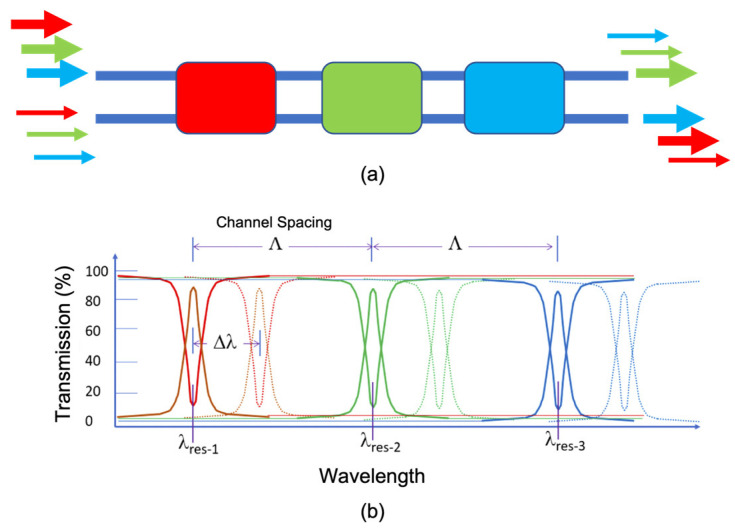
Block diagram of (**a**) the 2 × 2 × 3λ multicrossbar switch in a series connection of three resonant devices, and (**b**) the overall spectral response for the two states of each constituent.

**Figure 12 sensors-23-00626-f012:**
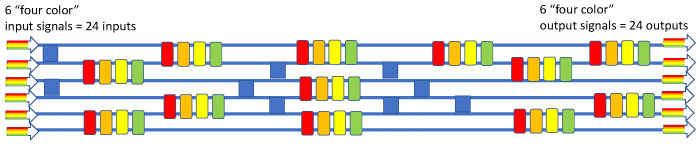
Block diagram of the 6 × 6 × 4λ wavelength-selective switch in six stages.

**Figure 13 sensors-23-00626-f013:**
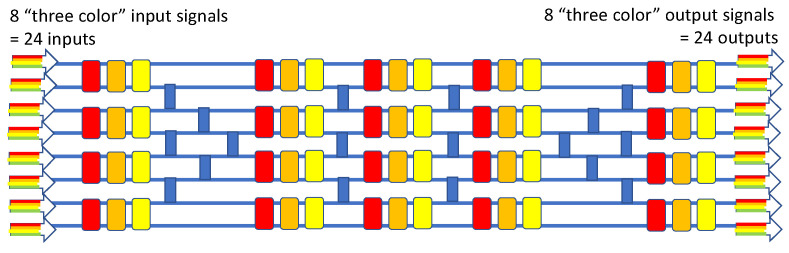
Block diagram of the 8 × 8 × 3λ wavelength-selective switch.

**Figure 14 sensors-23-00626-f014:**
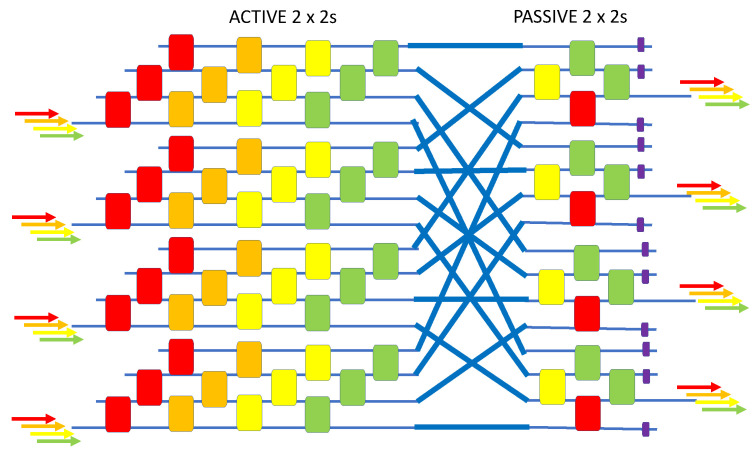
Block diagram of the 4 × 4 × 4λ wavelength cross-connect switch.

## Data Availability

Code and data available upon request to the corresponding author.

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
