# Peer review of "An Integrated Optical Circuit Architecture for Inverse-Designed Silicon Photonic Components"

_sensors, 2023, doi:10.3390/s23020626_

Round 1
Reviewer 1 Report
In this manuscript, the authors have proposed three inverse-designed, topologically optimized, integrated silicon-photonic devices: a 10×10 μm2 2×2 3-dB splitter/combiner, a matching 2×2 crossover, and a matching 2×2 all-forward add–drop resonator. They also present ten examples of these switching and computational functionalities in the context of WDM chips thanks to the unified shape, input-output location, and size that enable these IDCs to be organized in a simpler, parallel-waveguide circuit architecture. This manuscript is impressive and well-organized, but some unclear points must be addressed before publication.
1. The 2×2 3-dB splitter/combiner, 2×2 crossover, and 2×2 all-forward add–drop resonator presented in this section are designed with topology optimization. The performances of the three devices are only given by the top-view optical field profiles, as shown in Figures 1,2, and 3, respectively. I strongly suggest the authors add the spectra of these devices. Besides, I would like to see the comparisons in performance between these inverse-designed devices and the traditional forward-design devices.
2. The authors mentioned that the large spatial filter is important when designing a device with easy-to-fabricate large features. So, what are the spatial filters when designing these three devices? And what is the corresponding fabricating features? I think feature size is very critical for large-scale manufacturing.
3. The proposed 2×2 crossover is essentially a similar function as the waveguide crossing. Can the authors evaluate its advantages and disadvantages compared with the crossing? For me, the waveguide crossing is essentially symmetrical at the horizontal and vertical waveguides.
Author Response
In this manuscript, the authors have proposed three inverse-designed, topologically optimized, integrated silicon-photonic devices: a 10×10 μm2 2×2 3-dB splitter/combiner, a matching 2×2 crossover, and a matching 2×2 all-forward add–drop resonator. They also present ten examples of these switching and computational functionalities in the context of WDM chips thanks to the unified shape, input-output location, and size that enable these IDCs to be organized in a simpler, parallel-waveguide circuit architecture. This manuscript is impressive and well-organized, but some unclear points must be addressed before publication.
1.1 The 2×2 3-dB splitter/combiner, 2×2 crossover, and 2×2 all-forward add–drop resonator presented in this section are designed with topology optimization. The performances of the three devices are only given by the top-view optical field profiles, as shown in Figures 1,2, and 3, respectively. I strongly suggest the authors add the spectra of these devices. Besides, I would like to see the comparisons in performance between these inverse-designed devices and the traditional forward-design devices.
We would first like to thank the reviewer for their insightful and constructive feedback on our work. We highly value their opinion and believe that our work has significantly improved from the suggested revisions.
The original article displayed the optical field profiles of each device (for each input/output condition), along with key performance metrics of insertion loss and crosstalk at the operating polarization and wavelength. These devices are not broadband, however, so we believe adding transmission spectra do not add new information to the work. Thus, we have chosen not to include them. Although these devices could be made broadband by adding extra conditions to the optimization’s objective function (i.e., maximizing transmission at multiple wavelengths), there would be a trade-off with peak performance, device size, or minimum feature size. For the scope of the devices presented in this work, we believe we have provided all the necessary design and performance specifications at the operating polarization and wavelength.
To provide further context to relevant works, we agree with the reviewer that comparisons should be made to traditional devices. The following text has been added to the revised Sec. 3.2:
The potential advantages and benefits of the proposed crossover and splitter/combiner devices can be seen by comparing these designs to the current art of “traditional” splitters and crossovers. This comparison can be made in four areas: (1) the spectral transmission bandpass, (2) the device footprint, (3) the insertion loss, and (4) the optical crosstalk. The traditional 2 x 2 3-dB splitters comprise the classical directional coupler (CDC) (two, identical, parallel strip waveguides with an identical S-bend waveguide at each of four ports), cascaded DCs, the MMI coupler, and the asymmetric coupler with four tapers [49]. The Fig.-1 inverse-designed splitter shares with the CDC a narrow bandpass of 4 nm for 0.15 dB of output-ports imbalance, but the inverse device has a much smaller footprint, and its IL and XT metrics are superior. The traditional crossover is generally an X-shaped structure with waveguides oriented at right angles to each other, and with several waveguiding shapes being practical [41]. Our analysis of the Fig.-3 crossover shows that the spectral bandpass (4 nm for 0.2 dB of IL imbalance) is smaller than that of traditional designs; however, the Fig.-3 design—whose footprint is much similar in area to the traditional designs—offers convenient parallelism of the four waveguides, unlike the traditional 90-degree orientation. The design offers IL and XT comparable to those of traditional crossings.
1.2. The authors mentioned that the large spatial filter is important when designing a device with easy-to-fabricate large features. So, what are the spatial filters when designing these three devices? And what is the corresponding fabricating features? I think feature size is very critical for large-scale manufacturing.
We agree with the reviewer that feature size is critical when it comes to large-scale manufacturing (and small-scale research). In the optimizations, we use a low-pass spatial filter that is commonly used in topology optimization. This filter smooths out features that are less than a specified length scale (400 nm in this work). Although this does not guarantee all features are above 400 nm, it ensures that most features are well above the length scales of modern nanofabrication facilities (typically around 100 nm). To make this clearer for the reader, we have revised this following paragraph in Sec. 3.1 of the revised article:
A large low-pass spatial filter (commonly applied to modern topologically optimized devices [48]) is applied to finalize a design that contains typically larger features that are fabricated more reliably. The filter used smooths out features with length scales less than 10 pixels (400 nm). In general terms, a smaller low-pass spatial filter produces designs with smaller, more difficult to fabricate features, but also allows for higher performance, as a larger design space can be explored in optimization. We carefully manage this tradeoff to avoid extremes.
1.3. The proposed 2×2 crossover is essentially a similar function as the waveguide crossing. Can the authors evaluate its advantages and disadvantages compared with the crossing? For me, the waveguide crossing is essentially symmetrical at the horizontal and vertical waveguides.
We agree with the reviewer that our crossover device has the same general functionality as conventional waveguide crossings. The key difference with our crossover device is that it fits the same structural form as our other devices and seamlessly integrates into the greater parallel-waveguide circuit architecture. As traditional waveguide crossings form perpendicular connections [41], some design work would still be required to route the inputs/outputs to the rest of the circuit, which may not be trivial. The evanescent waveguide coupler may be a more similar device to our proposed crossover, but it tends to be a longer structure. To make this clear to the reader, the following text has been added to the revised Sec. 3.2, as follows:
This functionality is perhaps more like an evanescent waveguide coupler than a traditional waveguide crossing, which features perpendicular inputs/outputs [41] that would not easily fit our parallel-waveguide circuit architecture. The inverse-designed crossover has an advantage over conventional evanescent waveguide couplers, as the complex (inverse) index-engineering creates new effective paths so that full coupling can occur for virtually any device length (including very short lengths).
Reviewer 2 Report
In this manuscript, unified design strategy is presented to create a compact collection of inverse-designed, topologically optimized, integrated silicon-photonic devices including 10x10μm2 2x2 3dB splitter/combiner, matching 2x2 crossover, and matching 2x2 all-forward add–drop resonator. The manuscript needs to be revised:
1. The literature survey is incomplete and there is a lack of recent relevant developments, and therefore, the introduction and references should be revised.
2. Is there any optimization algorithm and/or method for the inverse-design approach? Please clarify and give proper description regarding this issue (this algorithm may be different for each structure).
3. Please explain if the structures considered as polarization-insensitive.
4. The resolution of Figure 11b is poor.
5. The manuscript needs to be carefully edited. For example Line 337: "the other seven input signals. The small purple squares in Fig. represent absorbers or ...", Line 176: "0.26 dB) and low crosstalk (-16.8 dB and -28.2 dB."
6. For the splitter/combiner, waveguide crossover, and all-forward add–drop resonator, the physical parameters and their transfer functions should be given, and a block diagram for each structure can be useful.
Author Response
In this work, the authors proposed a fabrication variation predictor model based on deep convolutional neural networks, which is trained to perform end-to-end prediction of the geometry of planar silicon photonic devices after photolithography and dry etching given the geometry in the design file. It has been demonstrated that the proposed model is able to predict various types of fabrication variation such as over/under-etching, corner rounding, filling of narrow channels and holes, and washing away of small features. This study brings significance to the field as it proposes a novel data-driven way to bridge the gap between sophisticated designs and fabricable devices. However, in order to fully show the usefulness of this method and get this work published, the authors need to address the comments listed below:
2.1. The literature survey is incomplete and there is a lack of recent relevant developments, and therefore, the introduction and references should be revised.
We would first like to thank the reviewer for their insightful and constructive feedback on our work. We highly value their opinion and believe that our work has significantly improved from the suggested revisions.
We agree with the reviewer that more discussion should be made on recent developments, as it helps set the proper context and relevancy of our work. We have added the following text (with many new references to relevant works) to the revised introduction:
A survey of the photonics literature shows that inverse optical design is a research-and-development area that has emerged strongly during the past five years. This design literature reports both theoretical and experimental results. We can identify specific developments that are relevant to the work reported in this paper. These include the computational techniques of inverse design that are assisted by deep neural networks [11], deep learning [12,13], physics-informed machine learning [14], black box algorithms [15], integral-equation methods [16], and linkage-tree genetic algorithms [17]. Additional computational methods include photonic emulation [18], the deep-adjoint approach [19], phase-injected topology optimization [20], and boundary-integral methods [21].
Practical implementation of inverse-designed components in commercial silicon-photonics foundries has been discussed by authors who examined spatial process variations [22,23], structural integrity [24], foundry fabrication constraints [25], and 300-mm multi-project wafer fabrication [26]. Progress on inverse-design nanophotonic components has been reported in the relevant areas of on-chip microresonators [27], planar on-chip mode sorters [17], all-optical logic devices [28], fiber-to-chip metamaterial edge couplers [29], optical beam steerers [30], couplers for on-chip single-photon sources [31], polarization-splitter-rotators [32], and photonic arbitrary beam splitters [33]. On the network level, inverse design developments include neural networks [34], Stokes receivers [35], and integrated photonic circuits [36].
We have extended the foregoing prior art to the silicon-photonic inverse-design area.
2.2. Is there any optimization algorithm and/or method for the inverse-design approach? Please clarify and give proper description regarding this issue (this algorithm may be different for each structure).
In the original article, Secs. 2 and 3 explain the optimization algorithm and how it applies to each of the three presented devices. As the method is not our own (like the FDFD solver we use within optimization), we believe using relevant citations would be more effective than re-explaining the theory in this work. However, to add further detail and clarity to the revised article, we have revised the following paragraph in Sec. 3:
The optimizations are set up for each device to maximize a target objective function with respect to its permittivity distribution. Target objective functions are typically a maximizing of optical throughput for a given mode and wavelength from an input waveguide to an output waveguide. The inverse design region in the middle of each device is discretized into a matrix of 40-nm pixels that can each take a permittivity value of silicon’s (εr,Si = 12.1) or silica’s (εr,SiO2 = 2.07) based on the direction the optimizer takes in maximizing the objective function. To efficiently optimize the thousands of pixels in the design region, topology optimization makes use of the adjoint method to calculate the optimization gradient, as it only requires two simulations to be made per iteration to calculate. The first simulation is a typical simulation of the device; the second simulation swaps the input and output directions to make a “reverse simulation” to find the gradient of the objective function with respect to the set of design parameters (pixel permittivity distribution). With the gradient calculated, an L-BFGS-B optimizer [47] adjusts the permittivity distribution along the gradient by a predetermined step size towards a slightly better performing design. This procedure is repeated until the objective function can no longer be maximized (fully converged). The overall method has shown success in designing compact high-performance silicon-photonic devices and is easily adaptable to new devices with unique design elements and functionalities, such as ours.
2.3. Please explain if the structures considered as polarization-insensitive.
Like all photonic devices, ours are polarization sensitive. Although this sensitivity can be reduced through extra conditions in the optimization (e.g., to maximize the transmission for TE and TM modes), for straightforwardness, we have chosen not to include it in this work. We have now made this clear for the reader with the following text added to the revised Sec. 3.1:
Although the devices in this work are optimized only for TE0 (i.e., they are not polarization-insensitive), they can just as readily be re-optimized for TM0. For polarization-insensitive operation, both polarizations can be optimized simultaneously through an extra condition in the objective function; however, this would likely come at the expense of peak performance.
2.4. The resolution of Figure 11b is poor.
The resolution of Fig. 11b has now been improved.
2.5. The manuscript needs to be carefully edited. For example Line 337: "the other seven input signals. The small purple squares in Fig. represent absorbers or ...", Line 176: "0.26 dB) and low crosstalk (-16.8 dB and -28.2 dB."
Thank you to the reviewer for spotting these mistakes in formatting. Both mistakes have been corrected, and the rest of the revised article has been carefully reviewed to make sure no other mistakes exist.
2.6. For the splitter/combiner, waveguide crossover, and all-forward add–drop resonator, the physical parameters and their transfer functions should be given, and a block diagram for each structure can be useful.
We agree with the reviewer that block diagrams would be useful additions to help clarify the functionality of the devices in Sec. 3. Figures 1–4 have been updated with the corresponding block diagrams, and the supporting captions and text have been updated accordingly.
Reviewer 3 Report
Researchers provide a toolkit of silicon photonic devices that are topologically optimized and inverse-designed in this study, assembled in a "plug-and-play" manner to create a variety of photonic integrated circuits, both passive and active with a tiny footprint. A 2 x 2 3-dB splitter/combiner, a 2 x 2 waveguide crossover, and a 2 x 2 all-forward add-drop resonator are all included in the silicon-on-insulator 1550-nm toolset. Authors shows that through the use of the thermo-optical effect, phase-change cladding, or free-carrier injection, the resonator could be converted into a 2x2 electro-optical crossbar switch. The toolset of photonic devices was shown to permit the compact circuit to achieve low insertion loss and minimal crosstalk for each of the 10 circuits described in this study. Overall, the flow of the manuscript is appropriate, and the results supports the claims of the authors. I have the following comments:
*Introduction and the literature should be extended to involve the relevant studies.
*Mathematical model/theory should be discussed.
*Did the authors experimentally verify their results/any of the simulations?
*Current state of the art can be discussed to better reflect the potential advantages and benefits of the presented designs.
Author Response
Researchers provide a toolkit of silicon photonic devices that are topologically optimized and inverse-designed in this study, assembled in a "plug-and-play" manner to create a variety of photonic integrated circuits, both passive and active with a tiny footprint. A 2 x 2 3-dB splitter/combiner, a 2 x 2 waveguide crossover, and a 2 x 2 all-forward add-drop resonator are all included in the silicon-on-insulator 1550-nm toolset. Authors shows that through the use of the thermo-optical effect, phase-change cladding, or free-carrier injection, the resonator could be converted into a 2x2 electro-optical crossbar switch. The toolset of photonic devices was shown to permit the compact circuit to achieve low insertion loss and minimal crosstalk for each of the 10 circuits described in this study. Overall, the flow of the manuscript is appropriate, and the results supports the claims of the authors. I have the following comments:
3.1. Introduction and the literature should be extended to involve the relevant studies.
We would first like to thank the reviewer for their insightful and constructive feedback on our work. We highly value their opinion and believe that our work has significantly improved from the suggested revisions.
We agree with the reviewer that more discussion should be made on relevant studies. To satisfy this comment, we have added the following text (with many new citations) to the revised introduction:
A survey of the photonics literature shows that inverse optical design is a research-and-development area that has emerged strongly during the past five years. This design literature reports both theoretical and experimental results. We can identify specific developments that are relevant to the work reported in this paper. These include the computational techniques of inverse design that are assisted by deep neural networks [11], deep learning [12,13], physics-informed machine learning [14], black box algorithms [15], integral-equation methods [16], and linkage-tree genetic algorithms [17]. Additional computational methods include photonic emulation [18], the deep-adjoint approach [19], phase-injected topology optimization [20], and boundary-integral methods [21].
Practical implementation of inverse-designed components in commercial silicon-photonics foundries has been discussed by authors who examined spatial process variations [22,23], structural integrity [24], foundry fabrication constraints [25], and 300-mm multi-project wafer fabrication [26]. Progress on inverse-design nanophotonic components has been reported in the relevant areas of on-chip microresonators [27], planar on-chip mode sorters [17], all-optical logic devices [28], fiber-to-chip metamaterial edge couplers [29], optical beam steerers [30], couplers for on-chip single-photon sources [31], polarization-splitter-rotators [32], and photonic arbitrary beam splitters [33]. On the network level, inverse design developments include neural networks [34], Stokes receivers [35], and integrated photonic circuits [36].
We have extended the foregoing prior art to the silicon-photonic inverse-design area.
3.2. Mathematical model/theory should be discussed.
In the original article, Secs. 2 and 3 explain the optimization algorithm and how it applies to each of the three presented devices. As the method is not our own (like the FDFD solver we use within optimization), we believe leading the readers to the citations we include to relevant work is more effective and efficient. However, to add further detail and clarity to the revised article, we have revised the following paragraph in Sec. 3:
The optimizations are set up for each device to maximize a target objective function with respect to its permittivity distribution. Target objective functions are typically a maximizing of optical throughput for a given mode and wavelength from an input waveguide to an output waveguide. The inverse design region in the middle of each device is discretized into a matrix of 40-nm pixels that can each take a permittivity value of silicon’s (εr,Si = 12.1) or silica’s (εr,SiO2 = 2.07) based on the direction the optimizer takes in maximizing the objective function. To efficiently optimize the thousands of pixels in the design region, topology optimization makes use of the adjoint method to calculate the optimization gradient, as it only requires two simulations to be made per iteration to calculate. The first simulation is a typical simulation of the device; the second simulation swaps the input and output directions to make a “reverse simulation” to find the gradient of the objective function with respect to the set of design parameters (pixel permittivity distribution). With the gradient calculated, an L-BFGS-B optimizer [47] adjusts the permittivity distribution along the gradient by a predetermined step size towards a slightly better performing design. This procedure is repeated until the objective function can no longer be maximized (fully converged). The overall method has shown success in designing compact high-performance silicon-photonic devices and is easily adaptable to new devices with unique design elements and functionalities, such as ours.
3.3. Did the authors experimentally verify their results/any of the simulations?
We have not experimentally verified our simulation results for this work. Although we agree that experimental results would help to support the design and simulation work on the devices presented in this work, the current scope was to outline the working principle of the circuit architecture and how inverse designs are integrated within it. Without experimental results, we cannot guarantee the exact same numbers in performance, but we believe the general performance range would be the same (as is generally expected from numerical simulations using Maxwell’s solvers). In other words, we use simulation to verify the working principle rather than the final performance level.
3.4. Current state of the art can be discussed to better reflect the potential advantages and benefits of the presented designs.
To provide further context to relevant works, we agree with the reviewer that comparisons should be made to previously demonstrated devices. The following text has been added to the revised Sec. 3.2:
The potential advantages and benefits of the proposed crossover and splitter/combiner devices can be seen by comparing these designs to the current art of “traditional” splitters and crossovers. This comparison can be made in four areas: (1) the spectral transmission bandpass, (2) the device footprint, (3) the insertion loss, and (4) the optical crosstalk. The traditional 2 x 2 3-dB splitters comprise the classical directional coupler (CDC) (two, identical, parallel strip waveguides with an identical S-bend waveguide at each of four ports), cascaded DCs, the MMI coupler, and the asymmetric coupler with four tapers [49]. The Fig.-1 inverse-designed splitter shares with the CDC a narrow bandpass of 4 nm for 0.15 dB of output-ports imbalance, but the inverse device has a much smaller footprint, and its IL and XT metrics are superior. The traditional crossover is generally an X-shaped structure with waveguides oriented at right angles to each other, and with several waveguiding shapes being practical [41]. Our analysis of the Fig.-3 crossover shows that the spectral bandpass (4 nm for 0.2 dB of IL imbalance) is smaller than that of traditional designs; however, the Fig.-3 design—whose footprint is much similar in area to the traditional designs—offers convenient parallelism of the four waveguides, unlike the traditional 90-degree orientation. The design offers IL and XT comparable to those of traditional crossings.
Round 2
Reviewer 2 Report
The authors have tried to revised the manuscript. The revised version can be accepted for publication.